# The Role of Social Value Orientation in Chinese Adolescents’ Moral Emotion Attribution

**DOI:** 10.3390/bs13010003

**Published:** 2022-12-20

**Authors:** Zhanxing Li, Dong Dong, Jun Qiao

**Affiliations:** 1Institute of Social Psychology, School of Humanities and Social Sciences, Xi’an Jiaotong University, Xi’an 710049, China; 2Xi’an Tieyi Binhe School, Xi’an 710038, China

**Keywords:** social value orientation, moral emotion attribution, adolescents

## Abstract

Previous studies have explored the role of cognitive factors and sympathy in children’s development of moral emotion attribution, but the effect of personal dispositional factors on adolescents’ moral emotion expectancy has been neglected. In this study, we address this issue by testing adolescents’ moral emotion attribution with different social value orientation (SVO). Eight hundred and eighty Chinese adolescents were classified into proselfs, prosocials and mixed types in SVO and asked to indicate their moral emotions in four moral contexts (prosocial, antisocial, failing to act prosocially (FAP) and resisting antisocial impulse (RAI)). The findings revealed an obvious contextual effect in adolescents’ moral emotion attribution and the effect depends on SVO. Prosocials evaluated more positively than proselfs and mixed types in the prosocial and RAI contexts, but proselfs evaluated more positively than prosocials and mixed types in the antisocial and FAP contexts. The findings indicate that individual differences of adolescents’ moral emotion attribution have roots in their social value orientation, and suggest the role of dispositional factors in the processing of moral emotion.

## 1. Introduction

Moral emotion is regarded as an important component of human moral development, which is reflected in the emotions that individuals experience when they conduct morality-related behaviors, such as feeling guilty when hurting others or feeling happy when helping others. As an important motivational factor [1], moral emotion is robustly linked with moral identity [2] and plays a pivotal role in people’s prosocial and antisocial behaviors [3,4].

Previous studies have explored the development of children’s moral emotion with the task of moral emotion attributions [5,6]. In the task, children were presented with some stories involving transgressing context or prosocial context and asked to anticipate how the actors would feel. Children who ascribe negative emotions (e.g., guilt) to the norm transgressor or ascribe positive emotions (e.g., pride) to the altruists are considered to be showing more evolved moral motivation which reflect the internalization of moral rules and values. Researches indicate that most young children around the ages of 4–5 years expect the transgressors to be happy (the so-called happy-victimizer phenomenon) [5,6]. It’s not until 7 or 8 years of age that children ascribe negatively charged emotions as a consequence of moral transgressions.

Recent studies found that the happy victimizer reasoning (i.e., to reason that victimizers feel happy after transgressing others) is not an exclusive feature of childhood, but that it can also be detected in adolescents and adults [1,7,8], which suggests moral emotion attribution may not simply be an age-related phenomenon, but reflects individual differences in emotional processing. Moreover, an obvious contextual effect (or *actor effect*) was found in adolescents such that they tend to attribute more negative feelings to the transgressors in the antisocial context compared to the actors in the context of failing to act prosocially; in the same way, they tend to attribute more positive moral emotions to the prosocial actors in the prosocial context compared to the actors in the context of resisting antisocial impulses [9,10,11]. This contextual effect was found in both Chinese and Canadian adolescents [11,12].

Some cognitive factors and sympathy were revealed to contribute to the development of moral emotion attribution [13,14,15,16,17,18]. For example, a recent investigation [18] found that 4-year-old children who rated higher in sympathy were less likely to reason that the victimizer feels good, and that greater inhibitory control predicted faster decreases in happy victimizer reasoning from 4 to 6 years of age. However, few studies have considered the role of individual dispositional factors in adolescents’ moral emotion evaluations. It is proposed that moral emotion attributions might reflect important inter-individual differences in morally relevant personal dispositions because the association between moral emotion attributions and behaviors is not moderated by age during adolescence [3]. Moreover, some personality traits such as conscientiousness and agreeableness could predict the development of moral emotion attributions from childhood to middle adolescence [19].

While moral emotion attribution reflects the difference between a person’s concern for his own interests and others [5,6], previous studies did not directly examine the role of personality traits reflecting self- vs. other-preference, such as social value orientation (SVO). Social value orientation is regarded as an important personality factor because it is temporally reliable in the test-retest measurement and rarely affected by situations [20,21,22,23] and can be partly explained by other personality traits and differences in biological constitutions [24,25,26,27]. The concept is originated from the game theory exploring people’s social preferences in social dilemmas without mutual interdependence [20,28] and is defined as individuals’ preferences for a particular apportionment of unspecified outcomes between self and others [20,28,29,30]. In Messick and McClintock’s seminal work [28], the scholars used decomposed games in which subjects were presented with a series of two-player binary options to select between them. Five patterns of social value orientation are generally identified in previous studies—i.e., individualist, competitor, cooperator, egalitarian, altruist [20,29,30]. Individualists tend to maximize their own results regardless of the results of others. Competitors are inclined to maximize the relative differences between their own results and those of others. Cooperators are motivated to maximize the joint results of themselves and others. Egalitarians tend to minimize the absolute differences between their own and others’ results. Altruists seek to maximize the results of their peers or others. While cooperators, egalitarians, and altruists can be integrated into the “prosocial” orientation, individualists and competitors can be integrated into the “proself” orientation [21,22]. Research with Chinese adolescent samples [20] showed that the representations of egalitarians, cooperators, and altruists were relatively low, and that the proportions of competitors increased with age from 9 to 14 years old. The ratios of prosocials decreased from 11 to 14 years old, but increased after entering adulthood. The results suggest that with age children are more self-concerned during adolescence. According to theory about moral identity, the centrality of some moral values to self is a key source of moral identity [31], which is intimately connected with moral emotions [2]. SVO may affect adolescents’ moral emotional process in different ways with age, which can be detected in adolescents’ moral emotion attributions at different ages.

According to an emerging stream of literature, prosocials displayed more others- or public-oriented concerns than proselfs in many social behaviors, such as trusting [32,33], sharing [34,35,36], helping [37], cooperation [27,38,39] and pro-environmental behaviors [40,41,42], but few studies have focused on the role of SVO in processing moral emotion. Some study has explored the link between SVO and emotion, but mainly focused on the immediate emotional reactions rather than the anticipated emotions between prosocials and proselfs when faced with (im)moral events [43]. According to Tracy and Robins’s process model of self-conscious emotions [44], people often consider standards, rules, and social values which are congruent with one’s self in order to elicit moral emotions. Adolescents who consider moral values as central to their self will anticipate stronger emotional reactions when moral rules are violated or observed by themselves. Reasonably, prosocials will be more likely to feel happy than proselfs when they sacrifice themselves to benefit others or restrain improper impulses for the congruence of value orientation with behaviors. In contrast, proselfs put much emphasis on their own benefits, thus they will not have negative emotional experiences even if they have brought harm to others but will feel happy for getting what they wanted. Additionally, previous research has showed that children who attributed negative emotions to the victimizer often rationalize the victimizer’s responses by referencing to others’ welfare, or the possible harm the victim might face, while children who attribute positive emotions to the victimizer often consider the benefits the victimizer might receive [45,46,47], which also suggest the association between moral emotion attributions and self–other orientations. As direct evidence, one recent study showed that self-oriented justice sensitivity (caring more for justice for one’s self) predicts more positive emotion attributions onto social norm transgressors while other-oriented justice sensitivity (caring more for justice for others) predicts less positive emotion attributions [48]. Different concern for one’s self and others can be directly reflected in SVO which may have a carry-on effect on adolescents’ moral emotion attributions.

In sum, the current study explored the influence of SVO on adolescents’ moral emotion attribution in different situational contexts, with an intent to examine the role of personal dispositions in moral emotion processing. The study recruited Chinese students in junior high school, senior high school, and college students as subjects, constituting three age groups: early adolescence, middle adolescence, and late adolescence. China is a country typical of Eastern culture and collectivism. Some scholars have proposed that compared with Westerners, people from Eastern cultures are more likely to see a highly moral person as societally oriented [49]. Some research demonstrated that compared to proself individuals, prosocial individuals are more likely to endorse collectivism values [50]. The investigation of Chinese adolescents would complement the defects of previous studies that mainly focus on Western subjects. A Chinese version of the triple-dominance measure [20] was used to measure the adolescents’ SVO. Although there are other alternative measurements (e.g., the SVO slider measure), this measure has been proven to have high reliability and validity in China [20] and has been widely used in Chinese teenagers. In reference to previous studies [5,6,9,10,11], the vignettes-based method was used to examine adolescents’ moral emotion attribution in four contexts: prosocial, antisocial, failing to act prosocially (FAP), and resisting antisocial impulse (RAI). The prosocial context depicted a person who sacrifices their own interests to help others. The antisocial context depicted a person who violates moral norms to meet their own needs. The FAP context depicted a person who knows the difficulties of others but does not lend a helping hand. The RAI context depicted a person who plans to engage in antisocial behavior but suppresses the impulse. The hypothesis is that SVO would influence adolescents’ moral emotion attribution and that this would differ according to different moral contexts. To maintain an integral moral self, prosocials would attribute more positive emotions to the character in the prosocial and RAI contexts than the proselfs, but they would attribute more negative emotions than the proselfs in the antisocial and FAP contexts as dissociation of moral identity.

## 2. Materials and Methods

### 2.1. Participants

Participants comprised 366 early adolescents in grades 7 and 8, 218 middle adolescents in grades 10 and 11, and 328 late adolescents in first- and second-year university. They were selected from Xi’an, a large city in the northwest of China. Nineteen early adolescents, eight middle adolescents, and five late adolescents were excluded from the analysis because they provided invalid data, leaving us with a total sample size of 880. Most of the participants came from middle to high socioeconomic backgrounds. Detailed demographic information is shown in Table 1.

The early and middle adolescents were recruited from consenting schools of Xi’an Jiaotong University. Compensation equivalent to approximately 2000 CNY was given to the schools for their participation in the study. The adolescents’ parents/guardians also gave their signed informed consent for the students to take part. The college students were recruited from first-year Introductory Psychology courses at Xi’an Jiaotong University and received class credit for their participation, and they gave their signed informed consent. The study was approved by the scientific research ethics committee of Xi’an Jiaotong University.

### 2.2. Measures

*Social Value Orientation* (SVO). We used the Chinese version of the triple-dominance measure to measure adolescents’ SVO [20]. The measure was adapted from the decomposed games in Van Lange et al. [21], which consists of nine forced-option dilemmas. Participants were asked to make a choice between two specific combinations of points for themselves and for another hypothetical person. The larger the points each one receives, the higher the profit they will get. Table 2 lists the sample items and the categorization mode in the study. For example, A (4,7) vs. B (10,5) means that participants faced with two options: “A—You will get 4 points and your partner will get 7 points vs. B—you will get 10 points and your partner will get 5 points”. Competitors’ choice pattern will be BAB in the three pairs of options because they tended to maximize the differences of benefit between themselves and the other. Individualists’ choice pattern will be BBB because they tended to maximize their own benefits. Egalitarians’ choice pattern will be AAB because they tended to minimize the difference between their own and others’ benefits. Cooperators’ choice pattern will be BBA because they tended to maximize the joint outcome. Altruists’ choice pattern will be ABA because they tended to maximize others’ benefits. There are three groups of items totally in the measure, with three items in each group. Besides the first group of items displayed in Table 2, the second group reversed choices A and choices B then added 1 point to the number, respectively, and the third group reversed choices A and B then added 2 points to the number, respectively. When two of three groups are consistent with one SVO pattern, the SVO pattern would be established; otherwise, the SVO was deemed as mixed orientation. This referred to Li et al.’s study [20]. 

*Moral Emotion Attribution*. Previous research commonly used vignettes including contexts of donating, helping and stealing to investigate young children and adolescents’ moral emotion attributions [5,6,9,10,11]. Some studies also included plagiarizing vignettes because adolescents will be faced with such a moral situation [9,10,11]. This study selected eight vignettes from these previous studies [9,10,11], which constituted four types of moral contexts: prosocial, antisocial, failing to act prosocially (FAP), and resisting antisocial impulse (RAI). The prosocial context involved a donation vignette (a person donates the money intended to buy personal gifts to children in a disaster area) and a helping vignette (a person brings an injured classmate to the hospital at the risk of being late for class). The antisocial context involved a stealing vignette (a person intentionally takes a toy from a store without paying for it) and a plagiarizing vignette (a person plagiarizes a classmate’s answers in an exam and gets a good score). The FAP context involved a non-donation vignette (a person does not donate money to children in need and instead buys gifts for themself) and a non-helping vignette (a person fears being late for class and thus does not help their injured classmates). The RAI context involved a non-stealing vignette (a person resists the impulse to steal a toy without paying for it and leaves the store empty-handed) and a non-plagiarizing vignette (a person resists the impulse to plagiarize from others and fails to achieve a good score). The eight vignettes were presented in randomized order. The action vignettes and the inaction vignettes were strictly parallel. For example, for the vignette describing a person donates the money to other children (action), there was a parallel vignette depicting the same situation characteristic that does not donate the money (inaction). Such a method of using hypothetical vignettes to assess moral emotion attributions has been widely used in previous studies [5,6,7,8,9,10,11].

### 2.3. Procedure

Participants completed a paper-and-pencil test comprising the triple-dominance measure and the eight moral vignettes about moral emotion attribution during a period of 15–20 min. Half of the participants were asked to complete the triple-dominance measure first and then to read and respond to the eight moral vignettes and the other half of participants reversed the order. For the triple-dominance measure, participants were asked to select between Option A and Option B for each dilemma. For the eight moral vignettes, participants were asked to imagine themselves as the protagonists in the vignettes and rate their overall feelings after acting as described on a six-point Likert type scale (1 = very unhappy to 6 = very happy). This follows the research of Krettenauer and Johnston [10].

## 3. Results

### 3.1. Distribution of Social Value Orientation

Through descriptive analysis, the distribution of adolescents’ SVO was shown in Table 3. Generally speaking, with the increase of age, the ratios of the competitors and individualists gradually rises (χ^2^(2) = 2.92, *p* = 0.232; χ^2^(2) = 105.08, *p* < 0.001). Meanwhile, the ratios of egalitarians and mixed social value orientations declines with age (χ^2^(2) = 7.88, *p* = 0.019; χ^2^(2) = 73.84, *p* < 0.001). Across all three age groups, the ratios of egalitarians, cooperators, and altruists were relatively low. Thus, egalitarians, cooperators, and altruists were integrated into an overall “prosocial” group. The individualists’ ratio was also low in early adolescents. They were combined with the ratios of competitors to form an overall “proself” group. The number of participants who fell into the prosocial group, the proself group, and the mixed group is shown in Figure 1. A chi-squared test showed that proselfs significantly increased with age (χ^2^(2) = 96.16, *p* < 0.001), but prosocials and mixed adolescents decreased significantly with age (χ^2^(2) = 12.12, *p* = 0.002; χ^2^(2) = 73.84, *p* < 0.001). These results replicate previous research findings with Chinese adolescents [20].

### 3.2. Variation of Moral Emotion Attributions in Participants with Different SVOs

Correlation analysis showed that ratings on moral emotions were significantly and positively correlated between the two vignettes in each context (ps ≤ 0.001), therefore, moral emotion attributions were calculated as the average rating across the two vignettes. Preliminary analysis showed that there was no significant order effect for emotion ratings in each moral context (ps ≥ 0.06). In addition, there was no significant gender effect on moral emotion attributions, thus the data for the different genders were combined for further analysis. The descriptive results are shown in Table 4. 

With the dependent variable moral emotion attributions, we conducted a repeated-measures analysis of variance (ANOVA) with SVO (prosocial, proself, mixed) as the between-subject variable and moral context (prosocial, antisocial, FAA, RAI) as the within-subject variable. We also added age as the covariate given the covariant relationship between age and SVO. The results revealed a significant main effect of context: F(3, 2619) = 144.71, MSE = 0.33, *p* < 0.001, η_p_^2^ = 0.14. Pairwise comparisons (Bonferroni) showed that ratings differed significantly among each context, with participants’ ratings highest in the prosocial context, followed by the RAI context and then the FAP context. Participants’ ratings were lowest in the antisocial context. The main effect of SVO was revealed as being significant—F(2, 873) = 3.07, MSE = 0.34, *p* = 0.047, η_p_^2^ = 0.01—but this effect was qualified by a significant interaction between context and SVO: F(6, 2619) = 8.89, MSE = 0.16, *p* < 0.001, η_p_^2^ = 0.02. Simple effect analysis indicated that the contextual effect existed in all three patterns of SVO (ps ≤ 0.001). Regarding the SVO, the proselfs rated significantly lower than the prosocials and mixed types in the prosocial context as well as in the RAI context, (ps ≤ 0.001). However, the outcome was reversed in the antisocial and FAP contexts (ps < 0.001). That is, proselfs rated significantly higher than the prosocials and mixed types in these two contexts. No significant differences were found between the prosocials and the mixed types in either context (see Figure 2). The covariate effect of age was not significant: F(1, 873) = 0.18, MSE = 0.68, *p* = 0.673, η_p_^2^ = 0.00.

A significant interaction between context and age was also revealed: F(3, 2619) = 48.50, MSE = 0.33, *p* < 0.001, η_p_^2^ = 0.05. Within each age group, there were significant contextual differences in the ratings of emotion attribution (ps < 0.001), with the ratings highest in the prosocial context, followed by RAI and FAP, and lowest in the antisocial context. In the prosocial and RAI contexts, early adolescents rated significantly higher than late adolescents and middle adolescents (ps < 0.001). This outcome was reversed in the FAP context. That is, early adolescents in this context rated significantly lower than late adolescents and middle adolescents (ps < 0.001). In the antisocial context, late adolescents rated significantly higher than middle adolescents, who in turn rated significantly higher than early adolescents (ps < 0.001) (see Figure 3). No other interactive effects were found.

## 4. Discussion

The current study investigated the role of social value orientation in Chinese adolescents’ moral emotion attributions in different moral contexts. We discovered an obvious contextual effect in adolescents’ moral emotion attribution. They evaluated more positively in the prosocial context and in the RAI context, but evaluated more negatively in the antisocial context and in the FAP context. Importantly, we found the contextual effect depended on SVO in guiding adolescents’ moral emotion attributions. Prosocials evaluated more positively than proselfs and mixed types in the prosocial and RAI contexts, but proselfs evaluated more positively than prosocials and mixed types in the antisocial and FAP contexts. The results manifest that SVO would contribute to adolescents’ moral feelings in different contexts and might be a source of individual differences in adolescents’ moral personality development. 

The findings corroborated prior studies which showed that a marked contextual effect exists in adolescents’ moral emotion attributions. The adolescents in the current study reported experiencing stronger negatively charged emotions in the antisocial context compared to those in the RAI context. In contrast, they reported stronger positively charged emotions in the prosocial context than in the FAP context. While the antisocial behavior and the prosocial behavior represent the acts of commission (i.e., actions), the behaviors of RAI and FAP represent the acts of omission or inhibition (i.e., inactions). Therefore, the findings suggest that contexts with actions trigger stronger moral emotions compared to contexts with inactions. This is consistent with previous studies that showed obvious *actor effect* [10,11]. 

In addition, we found a significant interaction between context and age (though the age effect was not significant), such that early adolescents reported significantly higher ratings than late adolescents and middle adolescents in the prosocial and RAI contexts, but these ratings were reversed in the FAP and antisocial contexts. Generally, adolescents’ moral sense was shown to decrease with age. According to Kohlberg’s theory of moral cognition development stages, early adolescents (i.e., 12-year-olds in this study) are in the “conventional moral reasoning” period. They commonly behave differently according to the rules approved by the society and seek praise from authorities to show that they are “good children”, which is often associated with moral emotions (e.g., guilty when violating social norms). With the enhancement of self-independence after entering late adolescence, adolescents gradually break through the restrictions of social rules and care about their inner feelings, thus self-oriented moral sentiment emerge. The findings warn us that more attention should be paid to the development of moral affections in students in higher grades since the arrival of puberty can easily arouse polarized emotional reactions which may cause serious transgressing behaviors.

As hypothesized, our findings showed that SVO would influence adolescents’ moral emotion attribution and that it differs according to different moral contexts. Prosocials evaluated more positively than proselfs and mixed types in the prosocial and RAI contexts; however, proselfs evaluated more positively than prosocials and mixed types in the antisocial and FAP contexts. This held true after controlling age as a covariant. It can be seen that prosocials have stronger moral feelings but proselfs have stronger self-concerned emotional experiences in the study. This is related to the difference between the prosocials’ and proselfs’ value pursuit. For prosocials, they seek to maintain a consistent moral self with the internalized moral values such that they will experience negative emotions after transgressing others and positive emotions for favoring others. But for proselfs, they care more about own interests, thus they would feel unhappy when their desires were not satisfied and respond positively when they get what they want, even at the expense of others. 

The effect of SVO on moral emotion attribution can explain the differences between prosocials and proselfs in some social behaviors given that moral emotion attribution is commonly seen as an important index of moral motivation [1]. As a case in point, some research found that prosocials are more likely to reciprocate when their partners show trusting behavior to them, but proselfs reciprocate less [33]. This might be due to the different anticipations of emotional consequences following reciprocity. Since prosocials regard trust and cooperation as important moral rules which contribute to the establishment of self-identity, they will expect to feel ashamed or guilty after violating these rules. In order to avoid these negative emotions, they will be motivated to comply with social rules and reciprocate more. But for proselfs, they attach more importance to maximizing their own interests, thus will not expect negative emotions even knowing that less reciprocity will exploit others. Some research found that when proselfs were induced to feel ashamed they tended to cooperate with the partner, but they will not if there is no inducement of ashamed emotion [51]. Future study should detect the possible intermediate role of moral emotion attribution directly between SVO and relevant social behaviors.

Additionally, the findings support the idea that some personal dispositional factors such as SVO will affect adolescents’ moral emotion attributions given that many previous works mainly focused on the role of cognitive ability and sympathy [13,14,15,16,17,18]. In contrast with previous studies which showed that SVO could influence individuals’ social behaviors such as trusting, sharing, and cooperation [32,33,34,35,36,37,38,39,40,41,42], the current study demonstrated that SVO can also lead to different emotional consequences in different moral contexts. It should be noted that we measured subjects’ SVOs by asking them to make allocating decisions facing with many two-player binary options. Previous studies showed that moral emotion attributions can predict children’s decision-making behavior in economic games [52,53]. The current study suggests that the causal link between moral emotion attributions and decision-making processes may not be unidirectional. 

There are several limitations to this study, which may be resolved in future studies. First, SVO was assessed rather than manipulated. Although this is common practice, subjects’ self-reported responses may not fully correspond to their actual performances in reality. As such, the findings need to be verified in combination with subjects’ actual behaviors in future studies. Second, the findings rely solely upon cross-sectional data. Although cross-sectional design can help us understand the causal relationship between SVO and moral emotional attribution at a group level, it does not allow for the investigation of the time-lagged relationships between variables and changes of dynamic processes during adolescence. Future research may adopt the cross-sequential design and analyze the predicting role of social value orientation in changes of adolescents’ moral emotion attribution using the cross-lagged regression method, thereby obtaining more valuable findings. Third, this study recruited adolescents from Asia (China) as subjects. Considering that most previous studies have used Western children characterized as WEIRDs (Western, educated, industrialized, rich, democratic) [54], the sampling of this study may also be regarded as being of merit, but the credibility and universality of the findings still warrant support of cross-cultural research. A related issue is that this study asked subjects to make self-evaluative judgments of moral emotion. A recent study showed that Chinese adolescents tended to report stronger negatively charged other-evaluative emotions than Canadian adolescents when observing others engaging in antisocial behavior, and less positive emotions for prosocial behavior, but no such difference was found in self-evaluative emotions [12]. Future research should comprehensively use two methods of evaluation to further reveal the differences or similarities in association between SVO and moral emotion attribution.

Despite these limitations, the current study is notable in that study has rarely been conducted to investigate the role of personality traits in moral emotional processing in older children, especially with Asian samples. Hill and Roberts [55] have proposed the necessity of integrating morality development and personality development, as personality and moral identity are like ‘‘fellow travelers’’ along one’s pathway through life and experience a critical period of development during childhood and adolescence. This study illustrates the interaction between personality and morality from the perspective of the moral emotional process, and demonstrates that the moral personality traits such as SVO diverge at least from early adolescence and will guide the expression of moral emotions. The study will spur more effort to reveal the intrinsic mechanisms of moral personality establishment and development. In addition, the study will help us understand why the happy victimizer reasoning still exists in adolescents and adults. Although adolescents commonly possess a high cognitive ability and a high level of sympathy, different experiences (environmental, educational, etc.) will shape different social value orientations, which will influence their moral expressions in daily life. The findings support that more researches are needed to pay attention to the influence of personality factors on moral emotion processing in the midway of life. Finally, the study may have some implications for better carrying out school education, since both social values and moral affections are important socializing contents of middle- to high-school students. The findings of the study indicate that these two aspects are connected in the middle school stage thus we can improve teenagers’ moral feelings by shaping their social value orientation from an early age, which surely needs a considerable interventional program to achieve this target. 

## Figures and Tables

**Figure 1 behavsci-13-00003-f001:**
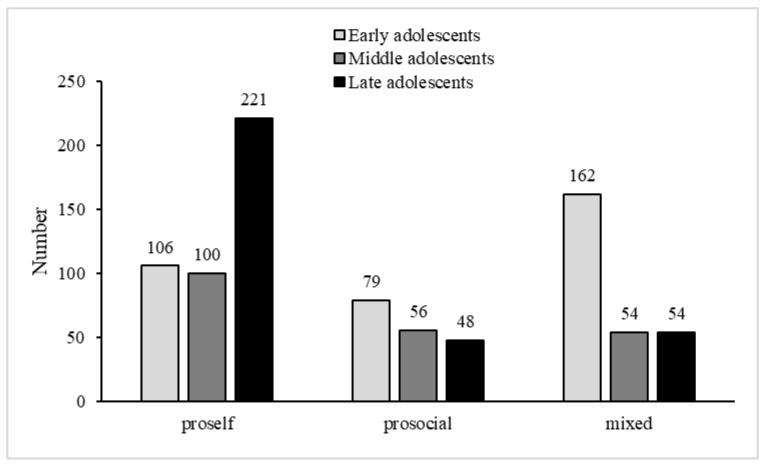
Number of Participants Falling into Three SVOs.

**Figure 2 behavsci-13-00003-f002:**
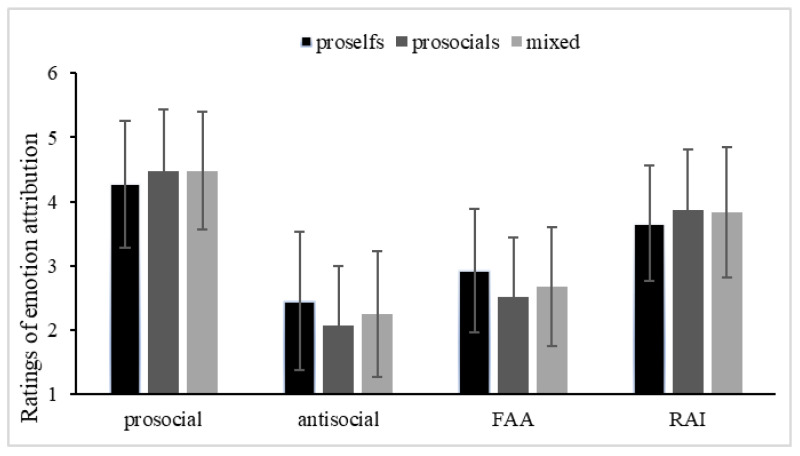
Ratings of Emotion Attribution in Different Contexts.

**Figure 3 behavsci-13-00003-f003:**
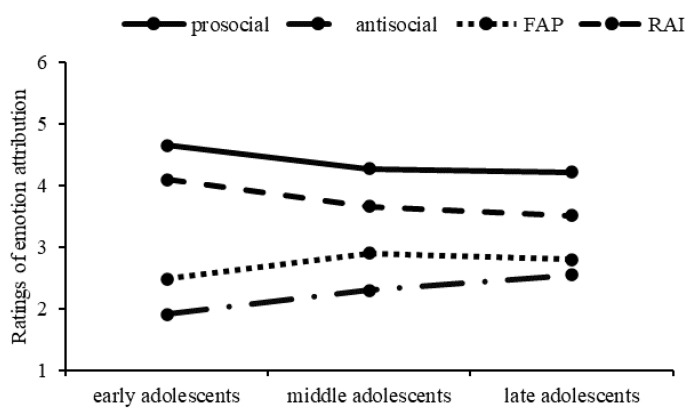
Ratings of Emotion Attribution in Three Age Groups.

**Table 1 behavsci-13-00003-t001:** Participant Demographic Information.

Group	Gender		Total	Mean Age (SD)
	Male	Female		
Early adolescents	197	150	347	12.81 (0.56)
Middle adolescents	130	78	210	15.59 (0.40)
Late adolescents	139	184	323	19.07 (1.87)

**Table 2 behavsci-13-00003-t002:** Sample Items and Categorization Mode of Social Value Orientation.

	A (4,7) vs. B (10,5)	A (5,4) vs. B (7,10)	A (5,10) vs. B (7,4)
Competitor	B	A	B
Individualist	B	B	B
Egalitarian	A	A	B
Cooperator	B	B	A
Altruist	A	B	A

**Table 3 behavsci-13-00003-t003:** Participants’ SVO Across Three Age Groups (%).

	Competitor	Individualist	Egalitarian	Cooperator	Altruist	Mixed
Early adolescents	25.4	5.2	15	2	5.8	46.7
Middle adolescents	27.6	20	12.9	10	3.8	25.7
Late adolescents	31.3	37.2	8	6.2	6	16.7

**Table 4 behavsci-13-00003-t004:** Means and Standard Deviations for Ratings of Moral Emotion in Participants According to SVOs.

	Early Adolescents				Middle Adolescents				Late Adolescents			
	Proselfs	Prosocials	Mixed	Total	Proselfs	Prosocials	Mixed	Total	Proselfs	Prosocials	Mixed	Total
Prosocial	4.44	4.73	4.78	4.66	4.23	4.30	4.28	4.26	4.08	4.38	4.17	4.14
	(1.13)	(0.99)	(0.86)	(0.99)	(1.06)	(0.95)	(0.84)	(0.97)	(0.86)	(0.90)	(0.97)	(0.89)
Antisocial	1.98	1.80	1.94	1.92	2.52	2.06	2.32	2.35	2.82	2.35	2.48	2.69
	(0.87)	(0.88)	(0.94)	(0.90)	(1.15)	(0.85)	(1.02)	(1.06)	(1.03)	(1.01)	(0.96)	(1.03)
FAP	2.73	2.22	2.49	2.50	3.06	2.76	2.87	2.93	3.04	2.59	2.75	2.93
	(1.06)	(0.95)	(0.97)	(1.01)	(0.98)	(0.90)	(0.80)	(0.92)	(0.88)	(0.88)	(0.85)	(0.89)
RAI	3.88	4.25	4.14	4.09	3.58	3.68	3.72	3.64	3.45	3.64	3.45	3.48
	(1.01)	(1.02)	(1.07)	(1.05)	(0.95)	(0.74)	(0.83)	(0.87)	(0.79)	(0.92)	(0.81)	(0.81)

Note: Standard Deviations in parentheses. FAP: failing to act prosocially; RAI: resisting antisocial impulse.

## Data Availability

The data presented in this study are available on request from the corresponding author. The data are not publicly available due to privacy reasons.

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
