# Peer review of "The Role of Social Value Orientation in Chinese Adolescents’ Moral Emotion Attribution"

_behavsci, 2022, doi:10.3390/bs13010003_

Round 1

Reviewer 1 Report

The present work is based on Impact of Social Value Orientation on Chinese Adolescents’ Moral Emotion Attribution.

-------------------------------------------------------

The Title is confuse, please try to modify it.

Lines 33-35: "Previous studies"? Which one? Please cite some studies that validate your sentence.

L39-40: "..More envolved __WITH__ moral emotions...." 

L40-42: "Research" or "Researches"? At the end of sentence, the author cite two references. So the correct is "Researches".

L55-59: The text is confuse and it seems that is correct to use the studies [11] and [12] as reference to all world. It is important to cite, but not comparing to all people.  In the sentence, it is interestin to change..

"IN SOME Cross-cultural studies, BASED ON A COMPARISON BETWEEN  CHINESE AND CANADIANS, THE AUTHORS SHOWN that this contextual effect WAS FOUND in both people..." Something like this.

L60-61: Which researches, please cite them in the sentence that you refered.

L70: Rewrite the sentence "..that children higher in symp..." Higher?

L71-72: Remove sentence..."These studies suggest that the development of moral emotion attribution has both cognitive and sympathetic foundations." It is void of significance. These what? Avoid to talk about past event. It is confuse in that way it was wrote.

L80: Please remove it. "To our knowledge.." What you mean? It is not correct to do auto-references, using personal pronouns even possessive ones.

L162: Remove "we" and re-write the sentence.

Final remarks:

The INTRODUCTION is too big, and is more similar to only a state-of-the-art. The introduction must to be improved to be "easier" to read. 

L171: "late adolescents in first- and second-year university". Adolescents  in the University? Are they really adolescents? Please, be clear on that. 19 years old isn't adolescent anymore.

L176: Please, put paragraph after "..Table 1."

L178: Please, explicite the currency. Is that, Yuan? Please, explicit the name of the currency.

L188: Explain more about SVO origin and definition. 

L198: Study or studies? The authors only citted 1 reference. Was the reference 11, the base of the Moral Emotion Attribution? Was it the standadized study of reference? Please put it clear in the text.

L231: Remove "we". Rewrite the sentence.

L240: Rewrite "...A chi square..." to "..A chi-squared..." in all paper, Figures, Tables as necessary.

L272: Table 3 shown a really tiny differences between the adolescents. What it means actually. It isn't hope that late adolescents present much higher values?

L326: Remove any personal word, such as, "we, our, I, us, me" when refering to the present papers' authors.

L348: Please, rewrite the paragraph. How to be sure, your research is the unique on the present scope? Did you read all researches around the world about it?  Is it better to say, that other similiar research was't found.

L351: Remove "our" and adapt the sentence.

Final remarks

Even the writing needs minor english reviews, mainly the introduction and references must to be reviewed. Most of the references are little old, 2 or more years ago. The introduction must to be improved, because it is too big, having no sections to improve the understanding.

I also miss to see some scater plots, for example, among the three different adolescents. It will show if there are any relations between too.

Author Response

The present work is based on Impact of Social Value Orientation on Chinese Adolescents’ Moral Emotion Attribution.

-------------------------------------------------------

The Title is confuse, please try to modify it.

Response: Thank you for your comment. We modified it into “The Role of Social Value Orientation in Chinese Adolescents’ Moral Emotion Attribution”.

Lines 33-35: "Previous studies"? Which one? Please cite some studies that validate your sentence.

Response: Thank you for your comment. We added relevant citations. Please see Line 36.

L39-40: "..More envolved __WITH__ moral emotions...." 

Response: Thank you for pointing this error. We revised this sentence as “Subjects who ascribe negative emotions to the norm transgressor (e.g., guilt) or ascribe positive emotions to the altruists are considered to be showing more evolved moral motivation which reflect the internalization of moral rules and values”. Please see Lines 40-43.

L40-42: "Research" or "Researches"? At the end of sentence, the author cite two references. So the correct is "Researches".

Response: Thank you for pointing this error. We corrected "Research" into "Researches". Please see Line 43.

L55-59: The text is confuse and it seems that is correct to use the studies [11] and [12] as reference to all world. It is important to cite, but not comparing to all people.  In the sentence, it is interesting to change.

"IN SOME Cross-cultural studies, BASED ON A COMPARISON BETWEEN CHINESE AND CANADIANS, THE AUTHORS SHOWN that this contextual effect WAS FOUND in both people..." Something like this.

Response: Thank you for your suggestion. We changed the sentence as “Some cross-cultural studies showed that this contextual effect was found in both Chinese and Canadians [11,12]”. Please see Lines 58-59.

L60-61: Which researches, please cite them in the sentence that you refered.

Response: Thank you for your comment. We cited relevant researches in the sentence. Please see Line 65.

L70: Rewrite the sentence "..that children higher in symp..." Higher?

Response: Thank you for your comment. We rewrited the sentence into “ A recent longitudinal investigation [15] found that children rated higher in sympathy were less likely to happy victimize at age 4”. Please see Lines 74-75.

L71-72: Remove sentence..."These studies suggest that the development of moral emotion attribution has both cognitive and sympathetic foundations." It is void of significance. These what? Avoid to talk about past event. It is confuse in that way it was wrote.

Response: Thank you for your suggestion. We removed the sentence as your suggestion.

L80: Please remove it. "To our knowledge.." What you mean? It is not correct to do auto-references, using personal pronouns even possessive ones.

Response: Thank you for your suggestion. We have removed "To our knowledge.." from this sentence.

L162: Remove "we" and re-write the sentence.

Response: Thank you for your comment. we removed “we” and re-writed the sentence as “The hypothesis is that SVO would influence adolescents’ moral emotion attribution and that this would differ according to different moral contexts.” Please see Lines 185-187.

Final remarks:

The INTRODUCTION is too big, and is more similar to only a state-of-the-art. The introduction must to be improved to be "easier" to read. 

Response: Thank you for your comment. We have compressed the introduction to make it more focused on the logic of this research topic. In addition, we added some content to make it easy for readers to understand. For example, we explained more about the origin and definition of SVO in the sixth paragraph, and we explained why SVO can be regarded as a personality trait in the seventh paragraph. Please see Lines 90-98 and Lines 111-117.

L171: "late adolescents in first- and second-year university". Adolescents in the University? Are they really adolescents? Please, be clear on that. 19 years old isn't adolescent anymore.

Response: Thank you for your comment. According to World Health Organization, Adolescence is the phase of life between childhood and adulthood, from ages 10 to 19 (see https://www.who.int/health-topics/adolescent-health#tab=tab_1). In this study, the average age of students in first- and second-year university is 19 years old, which conforms to the definition of the World Health Organization. In addition, many previous studies (for example, Jia et al., 2019) involved college students as the adolescent sample. The study is consistent with previous studies on the choice of late adolescents.

L176: Please, put paragraph after "..Table 1."

Response: Thank you for your suggestion. We have put paragraph after "..Table 1.’’

L178: Please, explicite the currency. Is that, Yuan? Please, explicit the name of the currency.

Response: Thank you for your comment. The name of the currency is Yuan. We have explicated it.

L188: Explain more about SVO origin and definition. 

Response: Thank you for your comment. We explained more about the SVO origin and definition in this paragraph, “The concept is originated from the Game Theory exploring people’s social preferences in social dilemma without mutual interdependence [20]. In Messick and McClintock’s seminal work [20], the scholars used decomposed games in which people are presented with a series of two-player binary options and asked to select. Social value orientation is defined as individuals’ preferences for a particular apportionment of unspecified outcomes between self and others.”

L198: Study or studies? The authors only cited 1 reference. Was the reference 11, the base of the Moral Emotion Attribution? Was it the standadized study of reference? Please put it clear in the text.

Response: Thank you for your comment. Actually, vignettes used in the previous studies are similar, basically including the contexts of stealing, sharing, helping, etc., both for young children and adolescents. Some studies also included plagiarizing vignettes because adolescents will face with such moral situation. This study recruited adolescents as sample, therefore, we refer to the vignettes from Krettenauer’s laboratory which mainly focused on adolescents’ moral emotion attribution. We added other references we cited in the studies, and put the reason clear in Line 254-258.

L231: Remove "we". Rewrite the sentence.

Response: Thank you for your comment. The sentence was rewrited as “Through descriptive analysis, the distribution of adolescents’ SVO was shown in Table 2.”  Please see Lines 293-294.

L240: Rewrite "...A chi square..." to "..A chi-squared..." in all paper, Figures, Tables as necessary.

Response: Thank you for pointing this error. We have rewrited "...A chi square..." to "..A chi-squared..." in full manuscript.

L272: Table 3 shown a really tiny differences between the adolescents. What it means actually. It isn't hope that late adolescents present much higher values?

Response: Thank you for your comment. Actually, there was no significant main effect of age in the study, but there was significant interaction between age and context. We added the total score of emotion rating of three age groups in different moral contexts (see Table 4). You can see that the total scores of emotion rating increase with age in the antisocial context and in the FAP context, but in the prosocial context and in the RAI context, the total scores of emotion rating decrease with age. In addition, we added a Figure 2 to show such trends more clearly.

L326: Remove any personal word, such as, "we, our, I, us, me" when refering to the present papers' authors.

Response: Thank you for your comment. We have removed the personal word such as "we, our, I, us, me" through the manuscript and rewrited the relevant sentences.

L348: Please, rewrite the paragraph. How to be sure, your research is the unique on the present scope? Did you read all researches around the world about it?  Is it better to say, that other similiar research was't found.

Response: Thank you for your comment. We changed the description as “Despite these limitations, the current study is notable in that other similar research was not found about the role of personality traits in moral emotion attribution in older children”. Please see Lines 423-426.

L351: Remove "our" and adapt the sentence.

Response: Thank you for your comment. We removed "our" from the sentence and changed the sentence into “The results contribute to the understanding of the interaction between morality and personality”.

Final remarks

Even the writing needs minor english reviews, mainly the introduction and references must to be reviewed. Most of the references are little old, 2 or more years ago. The introduction must to be improved, because it is too big, having no sections to improve the understanding.

I also miss to see some scater plots, for example, among the three different adolescents. It will show if there are any relations between too.

Response: Thank you for your comment. We feel it is too difficult to ensure that most of the references are within 2 years. We think it is proper to keep most of the references within 5 years. Therefore, we added several 5-year scope references. We have compressed the introduction and added some content to make it easy for readers to understand.

Actually, there were no significant relationships among the three age groups in terms of moral emotion attribution in four more contexts, ps > .05. We guess you might want to see the developmental trend of adolescents’ moral emotion attribution in different contexts. Therefore, a Figure 2 was added to display the results concisely.

Reviewer 2 Report

I appreciate this topic of study and applaud the authors for their grant and the sample size they were able to draw from. With that in mind, I find that the manuscript in its current state requires significant revisions as there is a lack of clarity and potentially an issue with the measures. Some concerns include:

1. In the abstract, the sympathetic mechanisms introduced is not conceptually clear. This could be a translation issue. Additionally, by scenario based situations, I believe you mean vignettes were utilized. Next, the variables are not clearly operationalized and at face value don't align well with the triple dominance measure literature.

2. Line 45 "happy victimizer reasoning" is not clear. What is meant by this phrase?

3. Personality factors does not appear to be measured by the vignettes. Variables used do not appear as constructs for individualistic, competitive, cooperative... perhaps the naming conventions are problematic? Antisocial for example does not align with individualistic nature. It has a different connotation.  As the data has already been collected, your best bet may be to clarify or even rename your variables and then expand on the limitations of your study.

4. I suggest you review the works of Mischkowski and Glockner (2016), Cameron et al. (2006), Garling et al., (2003), Bogaert et al. (2008), if you haven't done so already.

5. There is no mention of Eastern culture of which the sample is comprised and collectivism which should be considered when examining morality and prosocial behavior.

6. Overall, the literature review is not robust enough for the average reader. SVO needs to be explained more clearly; should explain its roots in Game Theory and Social Dilemma, etc. Triple dominance needs to be explained better as well.

Author Response

1. In the abstract, the sympathetic mechanisms introduced is not conceptually clear. This could be a translation issue. Additionally, by scenario based situations, I believe you mean vignettes were utilized. Next, the variables are not clearly operationalized and at face value don't align well with the triple dominance measure literature.

Responses: Thank you for your comment. We revised the first sentence to “Previous studies have explored the role of cognitive factors and sympathy in children’s development of moral emotion attribution” in the abstract. In addition, “Given a scenario-based method” was changed to “After reading some (im)moral vignettes”.

Many previous research used the triple dominance measure to assess SVO, but there are variations in the final classification, with some three patterns, some five patterns, or some more (see Murphy & Ackermann,2014). A common classification is to categorize the SVO into prosocials and proselfs. This study referred to Li et al. (2013)’s categorization, which finally classified SVO into three types—proselfs, prosocials and mixed. We added a description of the operational definition of this variable in the abstract to make it consistent with most relevant literature and easy for readers to understand. See line 13-15.

2. Line 45 "happy victimizer reasoning" is not clear. What is meant by this phrase?

Responses: Thank you for your comment. The “happy victimizer reasoning” echoes the happy victimizer phenomenon mentioned in the preceding paragraph. That means the tendency to reason victimizers to feel happy after transgressing others. To help readers understand this, we explained in parentheses. Please see Lines 47-48.

3. Personality factors does not appear to be measured by the vignettes. Variables used do not appear as constructs for individualistic, competitive, cooperative... perhaps the naming conventions are problematic? Antisocial for example does not align with individualistic nature. It has a different connotation. As the data has already been collected, your best bet may be to clarify or even rename your variables and then expand on the limitations of your study.

Responses: Thank you for your comment. Actually, the personality factors we investigated in this study was SVO. Extensive research regarding SVO suggests that it is a stable personality trait because it is temporally reliable in the test-retest measurement and rarely affected by situations [21,26], and associated with other personality characteristics such as Honesty-Humility trait [27]. Some scholars proposed that social value orientations can be partly accounted for by differences in biological constitutions [28] (see also in Bogaert et al., 2008, p456-458). Therefore, social value orientation reflected a stable individual difference in the way people evaluate outcomes for themselves and others [30]. We explained this in the seventh paragraph.

In this study, we classified SVOs into six types: individualist, competitor, cooperator, egalitarian, altruist and mixed, and combined individualists with competitors to form an overall “proself” group. The naming of five SVOs is consistent with previous studies (e.g., Li et al., 2013). The main purpose of this study is to investigate whether there are differences between prosocials and proselfs in the moral emotion attributions in different moral contexts. The positive emotions generated from antisocial behaviors is consistent with the essence of individualism because they both reflect the strong self-orientation.

4. I suggest you review the works of Mischkowski and Glockner (2016), Cameron et al. (2006), Garling et al., (2003), Bogaert et al. (2008), if you haven't done so already.

Responses: Thank you for your suggestion. We noted that these studies concerned about the relationship between SVO and cooperation and pro-environmental behavior. After reading them thoroughly, we cited them in the right place of the texts.

5. There is no mention of Eastern culture of which the sample is comprised and collectivism which should be considered when examining morality and prosocial behavior.

Responses: Thank you for your comment. We added the reasons why we chose the subjects of Eastern culture and collectivism, “China is a typical country of Eastern culture and collectivism. Some scholars have proposed that compared with Westerners, people from Eastern cultures are more likely to see a highly moral person as societally oriented [44]. Some research demonstrated that compared to proself individuals, prosocial individuals are more likely to endorse collectivism values [45]. The investigation of Chinese adolescents would complement the defects of previous studies that mainly focus on Western subjects.”. see Lines 169-174.

6. Overall, the literature review is not robust enough for the average reader. SVO needs to be explained more clearly; should explain its roots in Game Theory and Social Dilemma, etc. Triple dominance needs to be explained better as well.

Responses: Thank you for your comment. We supplemented the origin of SVO and defined SVO more clearly in paragraph 6. “The concept is originated from the Game Theory exploring people’s social preferences in social dilemma without mutual interdependence [20]. In Messick and McClintock’s seminal work [20], the scholars used decomposed games in which people are present-ed with a series of two-player binary options and asked to select. Social value orienta-tion is defined as individuals’ preferences for a particular apportionment of unspeci-fied outcomes between self and others.” Triple Dominance Measure was also described in detail of its composition and categorization mode. See Lines 230-248.

Reviewer 3 Report

The manuscript is a relevant trial to connect two variables.

The scientific quality is relatively good and we got a new piece in the building of knowledge.

However,  I need several relatively small clarifications.

1. Authors insist in the beginning and in the last sentence too, that SVO is a personality variable, personal disposition, inner psychological variable, however. Why? Or how?
It rather can be seen as a behavioral indicator. In any case, it is a variable considering interpersonal relationships (what I prefer to DO regarding interpersonal situations with OTHER). I would not prefer o call it "personality" (expect if we all area of attitudes, values, behavioral indicators... call personality).

2. Regarding table 3. I wonder why standard error and not standard deviation?  Also in the same table, can we analyse, compare results in three age groups? Means are different and can we conclude something from those differences?

3. Finally lines 191-193. The scale is adapted from the known scale with clear "philosophy". In the original scale, we have to choose the same amount of money when others receive more or less so it means that sometimes we are not satisfied with our gains but also want to have comparative advances. Or the third option is when we receive even more in absolute amount but not 10 times more than others, but only, say 50% more.

From such options, we can be categorized as competitive, individualistic, etc.  However, I do not understand what can be measured by example in lines 191-193. What does option A mean? That somebody wants to devaluate him/herself only that someone unknown gets a bit more than in option B? Maybe it is something connecting with cross-cultural differences when we take this sample into account? I do not know, but if yes, the authors need to introduce readers to cultural differences and explain the virtues of culture when someone rather chooses 4 than 10, just to improve the gain of others for 1 or 2?

Author Response

1. Authors insist in the beginning and in the last sentence too, that SVO is a personality variable, personal disposition, inner psychological variable, however. Why? Or how?

It rather can be seen as a behavioral indicator. In any case, it is a variable considering interpersonal relationships (what I prefer to DO regarding interpersonal situations with OTHER). I would not prefer to call it "personality" (expect if we all area of attitudes, values, behavioral indicators... call personality).

Responses: Thank you for your comment. Extensive research regarding SVO suggests that it is a stable personality trait be-cause it is temporally reliable in the test-retest measurement and rarely affected by sit-uations [21,26-27], and associated with other personality characteristics such as Hones-ty-Humility trait [28]. Some scholars proposed that social value orientations can be partly accounted for by differences in biological constitutions [29,30] (see also in Bogaert et al., 2008, p456-458). Therefore, social value orientation reflected a stable individual difference in the way people evaluate outcomes for themselves and others. We added explanation in Lines 111-117.

2. Regarding table 3. I wonder why standard error and not standard deviation? Also in the same table, can we analyse, compare results in three age groups? Means are different and can we conclude something from those differences?

Responses: Actually it is the standard deviation but not the standard error. Thank you for pointing this error. We have revised this error in Table 3.

Although means are different in three age groups, the repeated-measures analysis of variance (ANOVA) did not reveal a significant main effect of age. The result was consistent with previous studies with adolescents as sample [11]. We complemented this result in Lines 337-338-.

Although the main effect of age was nonsignificant, the interaction between context and age was significant. We added the total score of emotion rating of three age groups in different moral contexts (see Table 4). You can see that the total scores of emotion rating increase with age in the antisocial context and in the FAP context, but in the prosocial context and in the RAI context, the total scores of emotion rating decrease with age. In addition, we added a Figure 2 to show such trends more clearly.

3. Finally lines 191-193. The scale is adapted from the known scale with clear "philosophy". In the original scale, we have to choose the same amount of money when others receive more or less so it means that sometimes we are not satisfied with our gains but also want to have comparative advances. Or the third option is when we receive even more in absolute amount but not 10 times more than others, but only, say 50% more.

From such options, we can be categorized as competitive, individualistic, etc.  However, I do not understand what can be measured by example in lines 191-193. What does option A mean? That somebody wants to devaluate him/herself only that someone unknown gets a bit more than in option B? Maybe it is something connecting with cross-cultural differences when we take this sample into account? I do not know, but if yes, the authors need to introduce readers to cultural differences and explain the virtues of culture when someone rather chooses 4 than 10, just to improve the gain of others for 1 or 2?

Responses: Thank you for your comment. We added Table 2 listing the sample items in the first line and the categorization mode in the study. For example, A (4,7) vs. B (10,5) means that subjects faced with two options: “A – You will get 4 points and your partner will get 7 points vs. B – you will get 10 points and your partner will get 5 points”. Competitors’ choice pattern will be BAB in the three pairs of options because they tended to maximize the differences of benefit between themselves and other. Individualists’ choice pattern will be BBB because they tended to maximize their own benefits. Egalitarians’ choice pattern will be AAB because they tended to minimize the difference between their own and others’ benefits. Cooperators’ choice pattern will be BBA because they tended to maximize the joint outcome. Altruists’ choice pattern will be ABA because they tended to maximize others’ benefits. There are three groups of items totally in the measure with three items in each group. Besides the first group of items displayed in Table 2, the second group reversed choices A and choices B then added 1 point to the number respectively, and the third group reversed choices A and B then added 2 points to the number respectively. When two of three groups are consistent with one SVO pattern, the SVO pattern could be established; otherwise the SVO was classified as the mixed orientation. See more information in Lines 230-248.

Round 2

Reviewer 2 Report

Thank you for your responses and the revisions to your manuscript based on feedback. There are still translation/improper English grammar use throughout the article that can be addressed by an editor. Otherwise this has been much improved.

Author Response

Thank you for your responses and the revisions to your manuscript based on feedback. There are still translation/improper English grammar use throughout the article that can be addressed by an editor. Otherwise this has been much improved.

Response: Thank you for your comment. We realized that there must be some grammatical errors in the text because English is not our native language. We have invited an English professional to edit the language. We are willing to accept further language editing service of MDPI if it is necessary for this version of manuscript.
